# Artificial Intelligence to Predict the BRAF V595E Mutation in Canine Urinary Bladder Urothelial Carcinomas

**DOI:** 10.3390/ani13152404

**Published:** 2023-07-25

**Authors:** Leonore Küchler, Caroline Posthaus, Kathrin Jäger, Franco Guscetti, Louise van der Weyden, Wolf von Bomhard, Jarno M. Schmidt, Dima Farra, Heike Aupperle-Lellbach, Alexandra Kehl, Sven Rottenberg, Simone de Brot

**Affiliations:** 1Institute of Animal Pathology, Vetsuisse Faculty, University of Bern, 3012 Bern, Switzerland; caroline.posthaus@unibe.ch (C.P.); sven.rottenberg@unibe.ch (S.R.); 2Laboklin GmbH & Co. KG, 97688 Bad Kissingen, Germany; jaeger@laboklin.com (K.J.); aupperle@laboklin.com (H.A.-L.); kehl@laboklin.com (A.K.); 3Institute of Pathology, Department of Comparative Experimental Pathology, School of Medicine, Technical University of Munich, 81675 Munich, Germany; 4Institute of Veterinary Pathology, Vetsuisse Faculty, University of Zurich, 8057 Zurich, Switzerland; franco.guscetti@vetpath.uzh.ch; 5Wellcome Sanger Institute, Cambridge CB10 1SA, UK; lvdw@sanger.ac.uk; 6Synlab Vet Animal Pathology Munich, 81477 Munich, Germany; wolf.vonbomhard@synlab.com; 7Small Animal Clinic Hofheim, 65719 Hofheim, Germany; jarnoschmidt@googlemail.com; 8Veterinary Public Health Institute, Vetsuisse Faculty, University of Bern, 3012 Bern, Switzerland; dima.farra@unibe.ch; 9COMPATH, Vetsuisse Faculty, University of Bern, 3012 Bern, Switzerland; 10Bern Center for Precision Medicine, University of Bern, 3008 Bern, Switzerland

**Keywords:** urothelial carcinoma (UC), *BRAF*, artificial intelligence (AI), histology, canine, PCR

## Abstract

**Simple Summary:**

In canine urothelial carcinoma, the *BRAF* gene is frequently mutated (V595E). To detect this mutation, urine or tissue samples are currently tested by PCR. Recent advances in digital pathology and the power of artificial intelligence (AI) have opened up new possibilities for the detection of genetic alterations through AI histology. This new approach offers a wide range of new opportunities in the field of diagnostic and predictive tumour marker detection. The aim of this study was to test the efficacy of AI histology to predict the presence of the *BRAF* mutation in canine bladder carcinomas and to assess its intratumoral heterogeneity. This is the first study to utilise AI histology to predict *BRAF* mutational status in canine urothelial cell carcinomas.

**Abstract:**

In dogs, the *BRAF* mutation (V595E) is common in bladder and prostate cancer and represents a specific diagnostic marker. Recent advantages in artificial intelligence (AI) offer new opportunities in the field of tumour marker detection. While AI histology studies have been conducted in humans to detect *BRAF* mutation in cancer, comparable studies in animals are lacking. In this study, we used commercially available AI histology software to predict *BRAF* mutation in whole slide images (WSI) of bladder urothelial carcinomas (UC) stained with haematoxylin and eosin (HE), based on a training (*n* = 81) and a validation set (*n* = 96). Among 96 WSI, 57 showed identical PCR and AI-based *BRAF* predictions, resulting in a sensitivity of 58% and a specificity of 63%. The sensitivity increased substantially to 89% when excluding small or poor-quality tissue sections. Test reliability depended on tumour differentiation (*p* < 0.01), presence of inflammation (*p* < 0.01), slide quality (*p* < 0.02) and sample size (*p* < 0.02). Based on a small subset of cases with available adjacent non-neoplastic urothelium, AI was able to distinguish malignant from benign epithelium. This is the first study to demonstrate the use of AI histology to predict *BRAF* mutation status in canine UC. Despite certain limitations, the results highlight the potential of AI in predicting molecular alterations in routine tissue sections.

## 1. Introduction

Urothelial cell carcinoma (UC) is the most common malignant tumour affecting the canine lower urinary tract [1,2,3,4,5,6]. Since these tumours are not only highly invasive and metastatic but also typically located at the trigonum of the bladder, treatment options are limited, and the prognosis remains guarded to poor [4,6,7]. Scottish Terriers are at an extraordinarily increased risk for developing UC (>20-fold higher than other breeds), and Shetland sheepdogs, West Highland White Terriers and other terriers are also known to be predisposed [2]. Other risk factors for the development of UC include female sex and being spayed or neutered [2,3]. The increased risk for certain breeds to develop UC strongly suggests an underlying genetic basis [1,2,4,5,6,8,9]. In recent years, it has been shown that the V595E mutation in *BRAF*, the canine homolog to human *BRAF* V600E, occurs in up to 87% of canine urinary bladder carcinomas, and known high-risk dog breeds are particularly frequently affected [1,3,10,11,12]. The *BRAF* gene encodes the BRAF protein, which plays a key role in the MAP kinase/ERK signalling pathway and regulates important cellular functions such as cell growth, differentiation, proliferation, senescence, and apoptosis [13]. In humans, mutations in *BRAF* frequently occur in various tumour types, such as melanoma, thyroid carcinoma, and colorectal carcinoma, and it has been shown that the *BRAF* mutation can show inter- and intratumoral heterogeneity [1,14,15,16,17]. Very recently, it has been shown that in addition to the high prevalence of *BRAF* mutation in neoplasia of the lower urinary tract and the prostate of the dog, the *BRAF* V595E variant is frequent in canine papillary oral squamous cell carcinomas [18]. However, it is rarely identified (melanocytoma, peripheral nerve sheath tumour) or absent in other tumour types of dogs [10]. Of note, the *BRAF* mutation is rarely observed in human UC, although UC in dogs and the muscle-invasive form of UC in humans share many histopathological similarities [2,4,5,6,7,8,19,20,21,22,23]. The reasons behind this disparity have not yet been elucidated conclusively. Furthermore, the prognostic relevance of *BRAF* mutations in canine UC and the identification of driver genes in UC without *BRAF* mutations remain to be determined [24].

In dogs, detection of *BRAF* V595E mutation is typically carried out by PCR testing and is primarily used as a diagnostic marker to detect carcinomas of the urinary tract or prostate [25,26]. While PCR testing is considered highly specific, it is important to be aware of false negative PCR results depending on the quality and heterogeneity of the tested urine or tissue [25]. The test result also fails to provide spatial information in heterogeneous tumour tissue and benign adjacent tissue.

To address these limitations, digital pathology offers a promising solution. Computer-assisted analysis of whole slide images (WSI) allows us to efficiently evaluate cancer tissue sections in an objective, quantitative, and reproducible way. Moreover, it can assess complex features like spatial interactions, which are challenging to evaluate through routine light microscopy [27]. Artificial intelligence (AI) has proven to be extremely powerful for extracting and assessing quantitative information from digital histology [28]. Using AI as a tool to detect tissue markers and to classify tumours, e.g., based on morphologic characteristics like tumour grade, level of invasion, and cellular pleomorphism, has resulted in a rapid evolution and expansion of digital pathology [27]. AI can also be applied to detect molecular features, such as the presence of specific genetic alterations and their possible heterogeneity in tissue [29,30].

In veterinary science, the power of AI in pathology and imaging has recently gained a great deal of attention and continues to do so [31,32,33,34]. However, only a few animal studies in this field are currently available. In order to test the options of AI to predict specific molecular alterations on routine, HE-stained tissue sections, a specific genetic mutation, and the associated disease needs to be defined first. In the field of canine cancer, the most characterised and most frequently tested somatic mutations are *c-kit* in mast cell tumours and *BRAF* in bladder and prostate cancer [3,11,12,35,36,37]. The use of AI to predict *c-kit* mutation has been reported recently [38]. Considering the lack of any AI histology studies on *BRAF* in dogs and with the availability of a specific PCR test for this mutation, we selected this gene for testing the options of AI. In recent years, AI has been successfully used to predInict *BRAF* mutation status with high accuracy in human tumours, such as melanoma, thyroid carcinoma and colorectal carcinoma from radiological scans or WSIs [14,16,39,40,41,42,43]. The present study aimed to test AI histology to predict the presence of the *BRAF* V595E mutation in canine urinary bladder urothelial carcinomas.

## 2. Materials and Methods

### 2.1. Case Selection

The pathology archives of the Institute of Animal Pathology were searched for cases of primary carcinomas of the canine bladder. Additional cases were provided in collaboration with the Institute of Veterinary Pathology in Zurich, Switzerland; Laboklin Bad Kissingen, Germany; Synlab Vet Animal Pathology Munich, Munich, Germany and the Small Animal Clinic Hofheim, Hofheim, Germany. Of 157 dogs, 177 haematoxylin and eosin (HE) stained slides of tumour tissue and/or benign bladder tissue were available for analysis. For eight dogs, two or more WSI were selected per case, as the different slides contained either variable amounts of neoplastic and benign tissue or differed significantly in tumour histomorphology. All slides were scanned using the NanoZoomer S360MD Slide scanner system (Hamamatsu Photonics, Shizuoka, Japan). Signalment data were obtained and included breed, sex, neutering status, and age. All cases were reviewed and classified by a single pathologist as ‘highly valuable’ (defined as transmural bladder sections with well-preserved histomorphology), ‘standard’ (non-transmural section with well-preserved histomorphology), or ‘poor’ (small sample size or ill-preserved tissue). Additionally, the tumours were classified according to their morphology (‘conventional UC’: predominantly urothelial differentiation; versus ‘non-conventional’: urothelial differentiation minor or absent); invasion (‘non-invasive’: tumour borders little or non-invasive; versus ‘invasive’: clear tumour infiltration into lamina propria or lamina muscularis), and associated inflammation (‘inflammatory’: moderate to large numbers of leukocytes infiltrate the tumour parenchyma focally, multifocally, or diffusely; versus ‘non-inflammatory’: none or low numbers of infiltrating leukocytes).

### 2.2. PCR

Extraction of DNA from the paraffin-embedded formalin-fixed (FFPE) samples was performed using the QIAamp^®^ DNA FFPE Tissue Kit (Qiagen, Hilden, Germany) according to the manufacturer’s instructions. Isolated DNA was examined for the presence of the *BRAF* mutation c.1784T > A by digital droplet polymerase chain reaction (ddPCR) using a mutation-specific TaqMan^®^ assay as described by Mochizuki et al. [25]. Analysis was performed using DropletReader (Bio-Rad, Feldkirchen, Germany) and QuantaSoft™ Software (Bio-Rad, Feldkirchen, Germany).

### 2.3. AI Histology

For the digital histological analysis, commercially available software was used (Visiopharm 2022.11, Hørsholm, Denmark). All histological analyses, as well as deep learning training, were based on the WSI of HE stained tissue sections on glass slides. The slides were assessed in the following workflow using three different automated analytic programs designed specifically for this project (Figure 1).
(i)Tissue detection.

Based on training slides (*n* = 8), decision forest classification at magnification 0.5× was used to detect any HE-stained tissue on the slide and to outline the generated tissue label as a region of interest (ROI). Where needed, manual corrections were performed, which were minor in the majority of cases.
(ii)Tissue and tumour segmentation.

Based on training slides (*n* = 42) and 540 K iterations, deep learning (U-Net) classification at magnification 3× was used to separate tumour parenchyma from the stroma and non-tumour tissue and to outline the generated parenchyma label as ROI. On all slides, manual corrections were performed where needed, which were minor in non-invasive and major in highly invasive or poor-quality tumours. The area of this ROI was then measured in an automated way to evaluate the sample size.
(iii)*BRAF* mutation prediction

The case selection was first divided into a training and a validation set (Table 1). Assuming that the ‘high quality’ cases would be most valuable for AI training, these samples were preferred for the training set. Otherwise, the division into the two groups was performed randomly. The training set included cases of bladder UC from 73 dogs (81 slides) from 30 different breeds. The validation set consisted of 84 cases (96 slides) of bladder UC from 34 different breeds. Of these validation cases, 34 were classified as high quality, 19 as standard, and 43 as poor. Fifty were classified as predominantly urothelial and 46 as minor or absent-urothelial differentiated. Thirty-four tumours were categorised as infiltrative, 23 as little or non-infiltrative, and for 39 tumours, no classification could be achieved due to a small sample size or poor quality.

Based on the 81 training slides and 277 K iterations, deep learning (U-Net) classification at magnification 2× was used to predict *BRAF* mutation as either positive or negative. For the training, the entire tissue section was either labelled as positive or negative, depending on the corresponding PCR result. Based on the generated deep learning feature, we then used three different threshold classifications to predict *BRAF* mutation for each pixel in the previously defined tumour parenchyma ROI of validation cases:(1)Threshold 0.6: positive (≥0.6 probability), negative (≥0.6 probability) or uncertain (<0.6 positive and <0.6 negative);(2)Threshold 0.7: positive (≥0.7 probability), negative (≥0.7 probability) or uncertain (<0.7 positive and <0.7 negative);(3)Threshold 0.5: positive (≥0.5 probability) or negative (≥0.5 probability).

The analytic programs generated the following outputs for the validation set: area of assessed tumour parenchyma (mm^2^) (multiple tissue fragments were summarised); absolute (mm^2^) and relative (%) tumour area predicted as *BRAF* positive, negative and, for thresholds 0.7 and 0.6, uncertain (Figure 1). The final AI-*BRAF* prediction was defined as either positive or negative based on the predominant label (e.g., tumour labelled as 47% *BRAF* positive, 45% *BRAF* negative, 8% uncertain is overall predicted as AI-*BRAF* positive). In order to assess whether the AI-*BRAF* prediction was correct, the corresponding PCR result available for each tumour was used as a gold standard and compared with the AI values. If the AI-*BRAF* assessment and the corresponding PCR result did not match, the case was interpreted to be incorrect by AI-based *BRAF* prediction.

Once labelled, the slides were reviewed in order to detect any correlation between AI-based *BRAF* prediction and histomorphology.

In 15 cases, benign urothelium adjacent to the tumour was present on the WSI. For these cases, we subdivided the slide into benign only and tumour only and performed the AI analyses for these two areas separately.

### 2.4. Statistical Analysis

A two-sample *t*-test was performed to compare the sample size (i.e., assessed tissue area defined as the region of interest) with the correct or incorrect prediction of the AI tool. The chi-square test was used to test for associations between AI-based *BRAF* prediction and quality (poor, standard, excellent), morphology (conventional urothelial versus non-conventional differentiation), inflammation (present or absent), and invasion (present or absent). For both tests, *p* values < 0.05 were considered to indicate a statistically significant difference. Statistical analysis was performed with NCSS 2022 (v22.04, Kaysville, UT, USA).

## 3. Results

Histologic analysis of the validation set confirmed the presence of well-preserved (standard and high quality) tissue in 53 out of 96 (55%) bladder UC tissue slides (Table 1) examined. For a large proportion (34/53; 64%), tissue samples consisted of transmural bladder sections, which were defined as high-quality histology sections. The remaining slides were of reduced tissue quality, with the presence of tissue artefacts (most commonly tissue folds, suboptimal tissue fixation) or a very small sample size of a few mm^2^ or less. Tumour differentiation was conventional urothelial in approximately half of the cases (50/96), whereas the other half were defined as non-conventional UC based on histomorphology. Invasion into underlying lamina propria or tunica muscularis was evident in the majority (34/57; 60%) of assessable tumour slides, whereas the remaining cases were non- or poorly invasive with primarily pushing and smooth tumour borders. Stromal inflammation was present in 39 out of 95 (41%) assessable cases, characterised by mild to moderate, predominantly lymphoplasmacytic infiltrates.

After the initial histomorphological examination, the designed AI tool was used to predict *BRAF* mutation on all available (*n* = 96) WSI. The PCR testing identified the majority of cases as positive (69/96, 72%, versus 27/96, 28%). The three different (0.5, 0.6 and 0.7 deep learning classifier probability) threshold analyses for mutation prediction yielded different tissue areas predicted to be mutation positive, negative, or uncertain (for thresholds of 0.6 and 0.7). However, the final prediction of the entire WSI remained the same when running these three analytical programs. Thus, the three threshold analyses identically predicted the respective tumours as mutation-positive or negative overall, even though the relative positive vs. negative areas differed for each threshold.

When comparing the AI-based *BRAF* prediction of each WSI with the corresponding PCR result, 57 out of 96 (59%) showed identical results, which was interpreted as the correct prediction (Table 2). False positive predictions were observed in approximately one-third of the negative tested cases (10/27; 37%), while false negative predictions occurred in 42% of the positive cases (29/69). Overall, the specificity (SP) of the AI mutation prediction was 63%, and the sensitivity (SE) was 58%.

The sensitivity increased significantly when the WSI of reduced quality (i.e., ill-preserved histomorphology or small sample size) were excluded (Table 3 and Appendix A). The best AI prediction performance was achieved for the high-quality samples (*n* = 34) with a sensitivity of 89% and specificity of 43%. Sensitivity was reduced to 47% and 33% for standard (*n* = 19) and poor quality (*n* = 43) WSI, respectively. In contrast, specificity was higher in WSI of reduced quality, ranging from 69% (poor quality) to 75% (standard sample).

When comparing the performance of the AI prediction with tumour histomorphology, the level of urothelial differentiation was highly relevant (*p* < 0.01) (Table 4). WSI of conventional UC with evident urothelial morphology (*n* = 50) had a high sensitivity of 85%, which dropped to 21% in cases with ill-defined or absent urothelial differentiation (*n* = 46). The opposite was true when comparing the specificity, which was significantly higher in non-conventional (88%) compared to conventional (20%) UC. With regard to the level of tumour invasiveness, the AI prediction was highly sensitive (SE 84%) for non-invasive tumours and highly specific (SP 82%) for invasive UC. Sensitivity and specificity were, however, low for invasive (*n* = 34; SE 35%) and non-invasive (*n* = 23; SP 25%) tumours, respectively. The remaining cases were not assessable for the level of invasion due to their small size or reduced quality. Considering tumour inflammation, the AI test sensitivity was low (SE 29%) for inflamed UC (*n* = 39), whereas the opposite was true for those with associated inflammation (*n* = 56) (SE 80%). The specificity was higher (SP = 73%) for inflamed UC when compared to non-inflamed tumours (SP 56%). In one case, the level of inflammation was not assessable due to poor quality.

Independent of the histomorphological quality, sample size played a significant role in the AI test reliability (*p* < 0.02) (Table 4). This effect was most evident and beneficial when investigating *BRAF* mutation-positive cases where a larger sample size correlated with a higher level of correct prediction (*p* < 0.003) (Appendix A). In contrast, *BRAF*-negative tumours had a tendency for false positive AI predictions with increasing sample size (*p* > 0.3). Test reliability also depended significantly on tumour differentiation (*p* < 0.01), presence of inflammation (*p* < 0.01), slide quality (*p* < 0.02) and sample size (*p* < 0.02).

In addition to the relative and overall *BRAF* mutation prediction, the AI assessment enabled the visualisation of the intratumour mutation heterogeneity, which allowed the investigation of the correlation of *BRAF* mutation with tumour histomorphology. As seen above for the different tumour classes, positive *BRAF* mutation prediction was associated with tumour regions characterised by papillary growth, a smooth and pushing rather than invasive tumour front, and urothelial differentiation (Figure 2 and Figure 3). Histomorphological characteristics which tended to correlate with a negative *BRAF* mutation prediction were the following: solid tumour growth or divergent non-urothelial differentiation, invasive growth, pronounced tumour inflammation, and poor tissue quality, including artefacts due to squeezing, inadequate preservation, and tears (Figure 3). Tumour areas predicted to be uncertain if the mutation is present or absent were commonly observed at the interface of areas predicted as positive and negative and frequently showed a variety of different features of both the aforementioned groups.

In addition to the direct correlation of AI-based *BRAF* mutation prediction and tumour histomorphology, the spatial visualisation of mutation prediction was also valuable for the comparative investigation of benign and neoplastic urothelium. For *n* = 15 WSI of the validation set, benign urothelium adjacent to the tumour was available. All but one case was PCR tested mutation positive. The majority (11/14; 79%) of these cases were correctly predicted to bear the mutation based on AI. In six of these PCR-confirmed and AI-predicted positive cases, AI classified benign urothelium as mutation-negative (Figure 4). One such case represented a conventional papillary UC from a Scottish Terrier, where the AI prediction correctly labelled the tumour as mutation-positive and furthermore classified small foci of dysplasia in the adjacent urothelium as positive, while morphologically unremarkable benign urothelium remained negative (Figure 3). The interface of the mutation-positive dysplastic foci and the mutation-negative benign urothelium was labelled as *BRAF*-uncertain. The ability of the AI tool to distinguish malignant from benign urothelium was confirmed in another case of *BRAF*-mutated UC of a Flat Coated Retriever, where the positive prediction was limited to neoplastic growth (Figure 5).

## 4. Discussion

Although AI and machine learning (ML) offer various promising advantages when compared to the routine histopathological examination of tumour tissue by a single pathologist, there are a number of significant challenges to their implementation in practice. In contrast to the defined and reproducible histomorphological parameters (e.g., level of invasion, cell or nuclear size, mitotic activity), the features which are relevant for AI-based decisions often remain unknown. This can pose a challenge to the interpretation and reproduction of results generated by AI. In the context of histology, it is key not to rely solely on AI results, but to consider them together with histomorphology. It is well known that the performance of AI models increases with the size and diversity of the training set. However, on occasions, the available dataset may only be relatively small, especially when dealing with rare tumours and/or subtypes. Another crucial factor is the division of the collection of WSIs into a training set and a validation set, and a large discrepancy between these two sets (such as an imbalance in the distribution of histological subtypes) can lead to poor results. In addition to the level of tissue section preparation, the quality of HE staining (staining depth, uniformity, and presence of dye impurities) and the imaging scan of the section are also factors affecting AI training and interpretation [44,45,46]. Therefore, the selection of one or multiple different slide scanners, scan resolution, and staining quality is relevant for AI-based histology studies [42,47,48,49]. For the present study, the AI tool was designed based on the conditions of a single institution (Institute of Animal Pathology, University of Bern), with one specific scanner (NanoZoomer S360MD, Hamamatsu Photonics), a defined (20×) scan resolution, and tissue slides with highly variable HE staining, histomorphology, sample size and quality. Studies using AI histology based on a single institution and the use of a single scanner have already been described in comparable human studies [50]. With respect to the prediction of the BRAF mutation in human cancer, various AI approaches have been described in the literature, and each has its strengths and weaknesses [14,17,43].

In our study, sample size, i.e., the area of neoplastic urothelium on the corresponding WSI, was confirmed to be important for the AI test performance. Large samples of mutation PCR-positive UC were often correctly predicted as such, whereas the opposite was true for mutation-negative cases with large sizes. When comparing the mean size of PCR BRAF-positive UC with their negative counterpart, it was apparent that mutated tumours were approximately twice as big in our validation set (mean *BRAF* mutation-positive: 16 mm^2^ versus mean *BRAF* negative: 9 mm^2^; *p* > 0.1). A possible explanation for the larger average size of *BRAF*-mutated tumours is a sampling bias. Alternatively, if this correlation is true that BRAF mutated tumours tend to be larger at the time they are biopsied, then their frequent papillary and poorly invasive growth most likely explains the larger size when compared to flat UC [2]. Being aware that sample size affects AI prediction, training sets need to be standardised for this criterium to increase the reliability of this or any other AI test. This was not done for this study and therefore represents a limitation, which needs to be addressed for future AI studies. A minor (not statistically significant) difference in sample size was indeed observed in the training set with larger tissue sections for mutated (261 mm^2^) compared to non-mutated UC (251 mm^2^). However, it is important to note that the sample size of training and validation slides cannot be compared directly as their tissue samples were defined differently. The sample of the training slide was defined as all available tissue, i.e., tumour parenchyma as well as stroma and surrounding non-neoplastic tissue, whereas the tissue sample for *BRAF* prediction was restricted to the tumour parenchyma of the UC. When comparing sample size with slide quality, it was evident that a larger sample size was associated with higher quality (with 31.93 mm^2^ for high quality, 6.57 mm^2^ for standard quality, and 2.37 mm^2^ for poor quality) as sample size was a criterion to define quality. We therefore, concluded that the AI possibly used the size of the tumour as a criterion for *BRAF* prediction, which would also explain why tumours of smaller size, like those of poor quality, were often misinterpreted by AI as negative when actually PCR BRAF positive while large PCR BRAF negative UC was frequently misinterpreted as positive. This correlation was to be expected since large specimens were usually transmural sections and therefore defined as high quality. Small samples were often squeezed, difficult to orient and/or had other artefacts that negatively affected the assessment and were therefore classified as poor quality. For future studies, it would be conceivable that training labels should be of similar size, independent of the size of the tissue section on the WSI. For this purpose, one or several regions with a specific size and shape (e.g., a square of 1 mm^2^) could be defined, and training labels created only in these regions.

Our study found that the level of tumour inflammation markedly affected the sensitivity of the AI-based *BRAF* prediction, which was low for inflamed UC and high in those without associated inflammation. In contrast, the specificity was higher in inflamed than non-inflamed UC. There was no considerable difference in the ratio of inflamed to non-inflamed UC when comparing the training group with the validation group or when comparing different mutation statuses. However, in contrast to the high quality of the UC within the training set, the inflamed UC in the validation set were also often of poor quality (21/39), which could explain the poor sensitivity of the AI-based *BRAF* prediction for this category.

Another factor we identified as being critical determinate for the accuracy of the AI test reliability was the level of urothelial differentiation. For UC with evident urothelial morphology (conventional UC), the mutation was detected by AI with a high level of sensitivity. However, the rate of false positives was high for these tumours. For UC with only subtle or absent urothelial morphology (non-conventional UC), the AI prediction was highly specific for the mutation-negative cases; however, the rate of false negative cases was high. The large discrepancy between sensitivity and specificity for non-conventional UC cases may have been influenced by the imbalance in the training set: 71/81 cases in the training showed conventional urothelial morphology and only 10 were of divergent differentiation. In addition, none of the latter were PCR negative for BRAF. Thus, the training set only included one non-conventional UC with positive BRAF PCR, which could explain the poor sensitivity of our AI tool for this category.

Similar to the extent of urothelial differentiation, the level of tumour invasiveness was closely associated with the AI test reliability. UC with a clearly invasive tumour border tended to be predicted as mutation negative on AI, which resulted in a high specificity but low sensitivity, whereas the non-invasive tumours had a high sensitivity and low specificity. The reason for the large discrepancy between the sensitivity and specificity of invasive and non-invasive UC cases was not obvious. However, 39 of 96 cases could not be assessed for their level of invasion due to sample size and/or quality. Thus, there were far fewer cases available for assessment of invasion than for other tumour characteristics, which limited the interpretation of this factor.

In human cancers with frequent (V600E) *BRAF* mutation, it is widely recognised that intratumoral heterogeneity exists; however, intratumoral *BRAF* mutation heterogeneity has not yet been described in canine cancer. In this study, we show that *BRAF* mutation prediction correlated with defined morphologic features. For example, areas with papillary growth, smooth and pushing tumour fronts and urothelial differentiation were often predicted as being *BRAF V595* positive (i.e., BRAF mutated). In contrast, areas with a more solid tumour growth pattern, divergent urothelial differentiation and invasive growth were often predicted as mutation negative. Pronounced tumour inflammation also led to *BRAF*-negative prediction. Additionally, artefacts like squeezing or poor tissue preservation were usually also interpreted to lack a *BRAF* mutation. Locations with ‘uncertain prediction’ were usually identified at the interface between positive and negative tumour areas. Thus, genetic heterogeneity within the tumour needs to be factored in when making a diagnosis, especially if the treatment involves therapeutic strategies that depend on the presence or absence of *BRAF* mutation. Indeed, studies have shown that *BRAF* mutation status influences the treatment response in human cancers [51,52], and there is evidence for a different therapeutic response when comparing *BRAF* mutated vs. non-mutated canine cancers [35,53,54].

The *BRAF* mutation is found in malignant urothelium and not benign bladder or in urine from healthy dogs [10,11,55], making it a highly-specific marker for UC, with additional advantages of being non-invasive and inexpensive if performed on urine. In cases where UC is suspected and the *BRAF* mutation is not detected by PCR in urine, a bladder biopsy is needed to confirm or exclude UC. Not uncommonly, histologic evaluation can be difficult when dealing with early forms of UC or only small endoscopic biopsies. In this situation, the pathologist needs to have a high level of experience to reach a definite diagnosis, and the risk of missing small cancerous lesions exists. The power of AI for supporting the pathologist in detecting early and small cancer foci has been demonstrated for bladder and other cancers in humans [56,57,58]. Even though limited to a small series (*n* = 14) of *BRAF*-mutated UC, the present study confirms that AI was able to distinguish between malignant and adjacent benign urothelium in six cases. Considering that the separation of benign and malignant tissue was not the main aim of this study, the training was not specifically set up to perform this task. Nevertheless, AI has shown that benign urothelium more closely resembles *BRAF* mutation negative rather than positive tumours, as benign regions were labelled as such in the present study. With optimised training and based on the promising results from human studies, it can be expected that AI will be able to reliably differentiate benign urothelium from neoplastic bladder tissue in dogs [59,60].

## 5. Conclusions

This is the first study to demonstrate the use of AI histology to predict *BRAF* mutation status in canine UC. Despite certain limitations, which we were able to define, the results highlight the potential of AI in predicting molecular alterations in routine tissue sections. Important potential confounding factors are sample size and quality, as well as tumour histomorphology. Once optimised for these features, AI is able to reliably predict *BRAF* mutation, detect intratumoral mutation heterogeneity and differentiate between malignant and benign urothelium.

## Figures and Tables

**Figure 1 animals-13-02404-f001:**
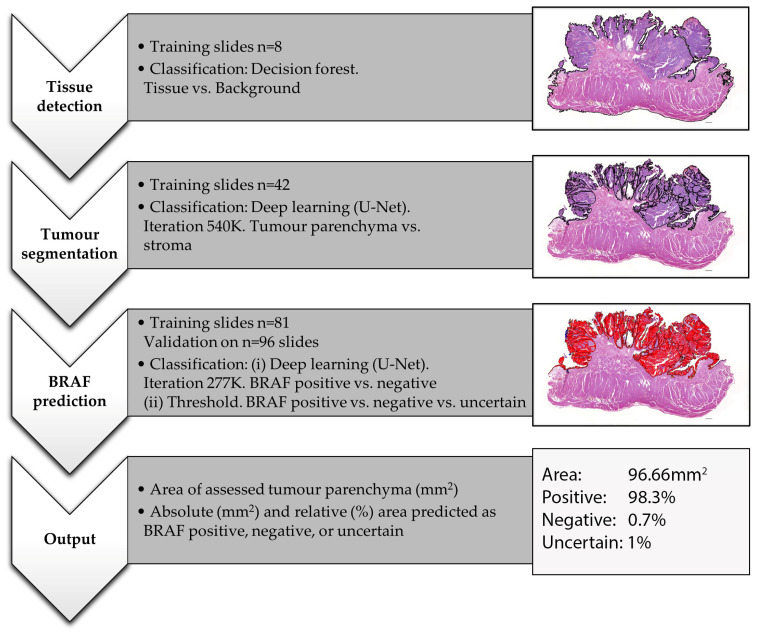
Workflow of the performed AI training and analysis for *BRAF* mutation prediction on WSI of HE-stained canine urothelial carcinomas. The images represent a case of bladder UC from an 11-year-old female neutered Jack Russell Terrier with positive BRAF PCR. This figure shows the output of threshold 0.6.

**Figure 2 animals-13-02404-f002:**
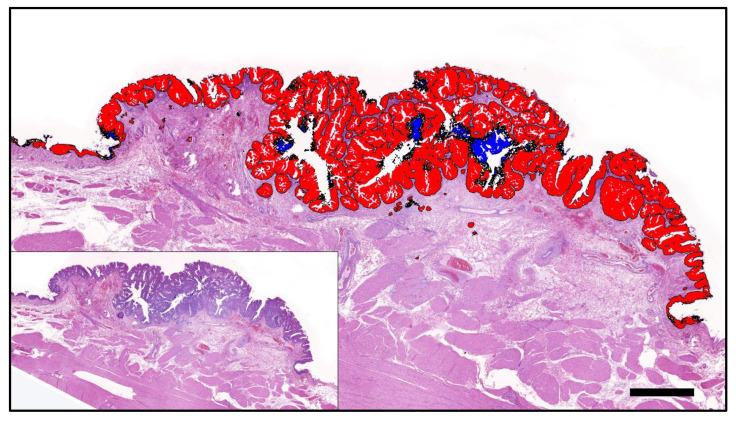
A papillary poorly invasive conventional urothelial carcinoma of a 9-year-old female neutered Scottish Terrier positive for *BRAF* mutation by PCR that was accurately predicted by AI histology. Labelling: red: *BRAF* positive; blue: *BRAF* negative; yellow: uncertain. Threshold classification 0.6. Bar indicates 1 mm.

**Figure 3 animals-13-02404-f003:**
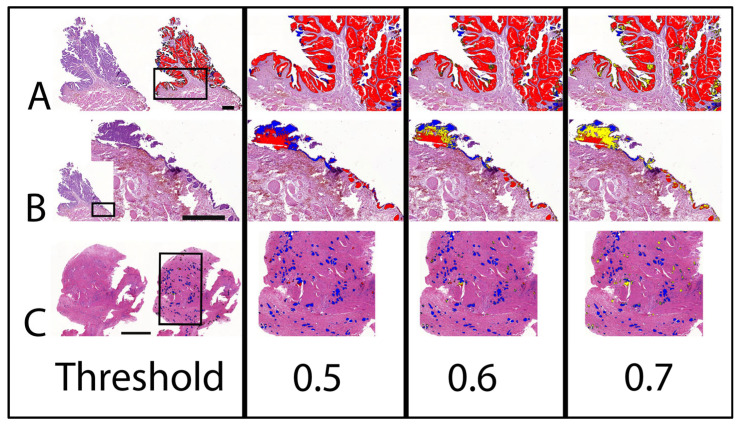
AI prediction of *BRAF* mutation on two cases of UC with different mutation status as confirmed by PCR. The labelling is indicated separately for the three different threshold classifications (0.5 to 0.7). Note that *BRAF* mutation negative labelling is changing to an uncertain label with an increasing threshold. (**A**) Non-invasive papillary UC with predominant positive labelling correctly predicted as mutation positive. Bar indicates 1 mm. (**B**) Adjacent benign tissue of (**A**). Bar indicates 0.5 mm. Benign flat urothelium is labelled negative, whereas Brunn nests are labelled positive. (**C**) Transmural highly invasive UC with predominant negative labelling correctly predicted as mutation negative. Bar indicates 1 mm. Labelling: red: *BRAF* positive; blue: *BRAF* negative; yellow: uncertain.

**Figure 4 animals-13-02404-f004:**
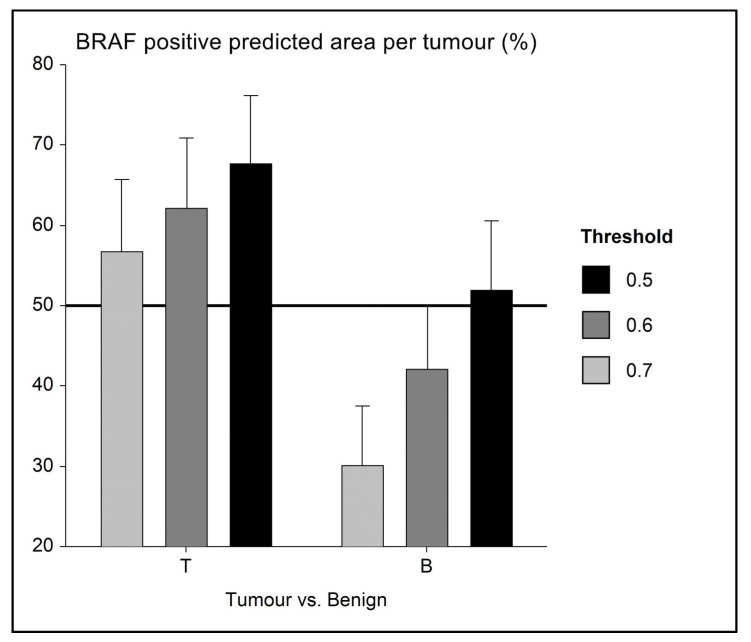
Error bar chart with columns representing mean group values and an indication of the standard error of the mean (SEM). In a small series (*n* = 14) of *BRAF*-mutated UC with adjacent benign urothelium, the level of mutation prediction (indicated as a percentage of *BRAF* positive labelling) differs, and AI is able to distinguish between tumour (*BRAF* positive) and benign (presumably *BRAF* negative) tissue. As a mean, benign tissue is correctly labelled as mutation negative (i.e., <50% BRAF positive) when using 0.6 and 0.7 AI threshold classifiers.

**Figure 5 animals-13-02404-f005:**
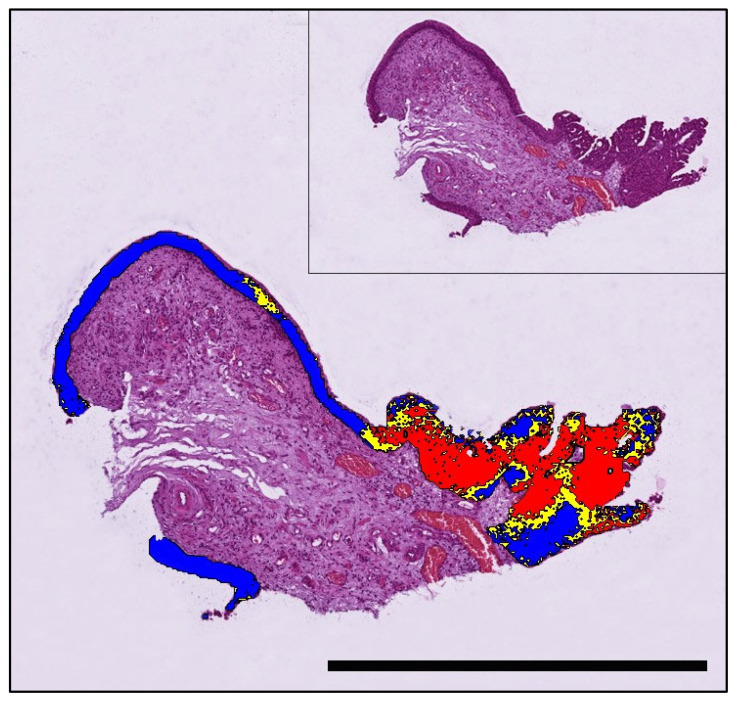
A papillary urothelial carcinoma of a 9-year-old Flat-coated Retriever that tested positive for *BRAF* mutation by PCR and was correctly predicted by AI histology. The mutation is predicted to be present in the neoplastic growth, whereas adjacent benign urothelium is labelled mutation-negative (threshold classification 0.6). Labelling: red: *BRAF* positive; blue: *BRAF* negative; yellow: uncertain. Bar indicates 1 mm.

**Table 1 animals-13-02404-t001:** Overview of the *n* = 177 WSI used for AI training and analysis. Abbreviations: conv.: conventional (urothelial); NA: not assessable; ROI: Region of interest (i.e., tissue area used for training or analysis).

	Training Set			Validation Set		
No. of slides	81			96		
PCR	positive	negative		positive	Negative	
	37	44		69	27	
Mean ROI	256 ± 135 mm^2^ (entire tissue section)			14 ± 3 mm^2^ (epithelium only)		
Quality	high	standard	poor	high	standard	poor
	80	1	0	34	19	43
PCR positive	37	0	0	27	15	27
PCR negative	43	1	0	7	4	16
Differentiation	conv.	non-conv.		conv.	non-conv.	
	71	10		50	46	
PCR positive	36	1		40	29	
PCR negative	35	9		10	17	
Invasion	present	absent	NA	present	Absent	NA
	44	36	1	34	23	39
PCR positive	22	15	0	23	19	27
PCR negative	22	21	1	11	4	12
Inflammation	present	absent	NA	present	Absent	NA
	32	49	0	39	56	1
PCR positive	15	22	0	28	40	1
PCR negative	17	27	0	11	16	0

**Table 2 animals-13-02404-t002:** Comparison of the AI-based *BRAF* prediction of each case with the corresponding PCR result.

AI Prediction	PCR Result: BRAF Positive	PCR Result: BRAF Negative	Total No. Cases
*BRAF* positive	40	10	50
*BRAF* negative	29	17	46
Total no. cases	69	27	96

**Table 3 animals-13-02404-t003:** Comparison of AI-based *BRAF* mutation prediction with confirmed PCR result based on specific histomorphological features. Shown in brackets is the number of cases where the PCR result matched the AI result. Abbreviations: SE: sensitivity; SP: specificity; stand: standard.

AI Prediction	No. Cases			
Quality	High	Stand		Poor
*BRAF* positive	28 (24)	8 (7)		14 (9)
*BRAF* negative	6 (3)	11 (3)		29 (11)
	SE [%]: 89	SE [%]: 47		SE [%]: 33
	SP [%]: 43	SP [%]: 75		SP [%]: 69
Urothelial differentiation	Present		Absent	
*BRAF* positive	42 (34)		8 (6)	
*BRAF* negative	8 (2)		38 (15)	
	SE [%]: 85		SE [%]: 21	
	SP [%]: 20		SP [%]: 88	
Invasive tumour front	Present		Absent	
*BRAF* positive	10 (8)		19 (16)	
*BRAF* negative	24 (9)		4 (1)	
	SE [%]: 35		SE [%]:84	
	SP [%]: 82		SP [%]: 25	
Inflammation	Present		Absent	
*BRAF* positive	11 (8)		39 (32)	
*BRAF* negative	28 (8)		17 (9)	
	SE [%]: 29		SE [%]: 80	
	SP [%]: 73		SP [%]: 56	

**Table 4 animals-13-02404-t004:** Assessing the AI tool’s reliability for correct *BRAF* mutation prediction based on specific features across different sample cohorts. A Chi-square test was performed for categorical data (urothelial differentiation, inflammation, slide quality, invasive growth, and mutation status) and a two-sided *t*-test for continuous data (sample size). *p* values of <0.05 were considered significant (in bold).

Feature	All Samples	Standard and High-Quality Samples Only	High-Quality Samples Only
Urothelial differentiation	**<0.01**	>0.1	>0.9
Inflammation	**<0.01**	**<0.03**	>0.7
Slide quality	**<0.02**	NA	NA
Sample size (ROI)	**<0.02**	>0.08	>0.3
Invasive growth	>0.08	>0.3	>0.2
Mutation status (PCR)	>0.6	>0.1	**<0.02**

## Data Availability

The data that support the findings of this study are available from the corresponding authors, L.K. and S.d.B., upon reasonable request.

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
