# Peer review of "Artificial Intelligence to Predict the BRAF V595E Mutation in Canine Urinary Bladder Urothelial Carcinomas"

_animals, 2023, doi:10.3390/ani13152404_

Round 1

Reviewer 1 Report

1. In addition to the level of tissue section preparation, the quality of HE staining (staining depth, uniformity, and presence of dye impurities) and the imaging scan of the section are also factors affecting AI training and interpretation. The digital pictures of all slices in this experiment were scanned by the same device. Control groups with different resolutions or pixels should be added, or the influence of photos scanned by different devices on the interpretation should be analyzed. It is also necessary to evaluate sections with different levels of staining.

2.Line199, "Threshold 0.7:" missing a word "positive"

3. In the training process, the AI first identified the tumor region and then correlated it with the given "BRAF positive or not" to accumulate experience; Similarly, in the process of interpretation, AI still needs to identify the tumor area first, and by comparing experience, it gives a judgment of whether it is BRAF positive or not under a certain threshold. Therefore, for sections with poor preparation, poor staining, high degree of tumor invasion, fuzzy tissue boundaries, inflammatory cell infiltration, and blood stasis, there are certain influences and errors in learning and interpretation. Can the range of tumor tissue be first demarcated by immunohistochemistry to reduce the error of AI recognition, and then used as training? In addition, tumor location, grade, progression, and tissue composition should also be considered.

Author Response

We thank the reviewers for their comments. All comments of the four reviewers have been addressed and the manuscript has been changed accordingly. All performed changes are indicated by yellow highlight in the revised manuscript. The reviewers' comments are listed below, written in bold, followed by the authors' answers.

In addition to the level of tissue section preparation, the quality of HE staining (staining depth, uniformity, and presence of dye impurities) and the imaging scan of the section are also factors affecting AI training and interpretation.

Yes, we agree with the reviewer and confirm that tissue staining and the type and quality of scan affects AI performance. This is particularly important when establishing an AI tool which is intended to be used on scans of different file format, as in a multi-institutional setting. We have added some phrases in the discussion with corresponding references to support this [1–3](lines 392-396).

The digital pictures of all slices in this experiment were scanned by the same device. Control groups with different resolutions or pixels should be added, or the influence of photos scanned by different devices on the interpretation should be analyzed. It is also necessary to evaluate sections with different levels of staining.

Yes, we agree with the reviewer that the number and type of scanner as well as scan resolution and staining is relevant for AI-based histology studies. We have added some phrases in the discussion with corresponding references to support this[4–7].

(lines 396-402).

For the present study, the AI tool was designed based on the conditions of a single institution (Institute of Animal Pathology, University Bern), i.e. designed to be used on one specific scanner (NanoZoomer S360MD, Hamamatsu Photonics), a defined (20x) scan resolution, and on HE stained tissue slides with highly variable HE staining, histomorphology, sample size and quality. Given the marked sample variation concerning tissue quality, size and staining (as to be expected for this canine cancer), we decided not to add more variation by using different scan formats. Similar investigations based on a single institution (and scanner respectively) have previously been performed in comparable human studies. We acknowledge that a comparison of the AI tool’s performance on different scan formats would provide important insights about the effect of scanning quality. However, this was not the aim of the present study and would require significant additional financial and analytic efforts. We would however consider to perform such comparisons as part of a follow-up study, ideally in a multi-institutional setting and with a larger set of cases.

It is also necessary to evaluate sections with different levels of staining.

Yes, we agree with the reviewer that staining variation is highly relevant when it comes to AI histology. As indicated above, this has been clarified in the manuscript and relevant references have been added.

The present slide collection was defined by a large variation in staining, with a wide range of staining intensity as well as variable eosin/haematoxylin balance. We can therefore confirm that AI training as well as analysis were performed on slides with variable staining levels. This situation also reflects the routine diagnostic setting where staining variation of canine bladder cancer tissue samples is common.  For a specific comparison of different staining features on AI performance, serial tissue sections with defined staining variation would be required. Such staining comparisons would enable to define the relevance of staining quality on AI performance, but would require significant extra analytic and financial investments and are outside the aim of the present study. We do however acknowledge that there is a yet unmet but urgent need to investigate the role of HE staining quality on this and other AI-based histologic analysis.

  1. Line199, "Threshold 0.7:" missing a word "positive"

This has been added (line 201).

In the training process, the AI first identified the tumor region and then correlated it with the given "BRAF positive or not" to accumulate experience; Similarly, in the process of interpretation, AI still needs to identify the tumor area first, and by comparing experience, it gives a judgment of whether it is BRAF positive or not under a certain threshold. Therefore, for sections with poor preparation, poor staining, high degree of tumor invasion, fuzzy tissue boundaries, inflammatory cell infiltration, and blood stasis, there are certain influences and errors in learning and interpretation. Can the range of tumor tissue be first demarcated by immunohistochemistry to reduce the error of AI recognition, and then used as training?

We can confirm that the AI training was performed on non-segmented tissue sections, which contained variable amounts of tumour and non-neoplastic bladder tissue and which were overall labelled as either BRAF positive or negative. By using whole-slide training annotations, this approach corresponds to a semi-supervised deep learning method which has been demonstrated to be effective and robust for AI histology [8–10]. Consequent AI analysis was only run on defined tumour parenchyma as BRAF mutation status was considered to be primarily relevant for neoplastic epithelium. We appreciate the reviewer’s comment about using IHC (e.g. cytokeratin) to reliably and non-manually distinguish tumour from benign tissue. We agree that this method would be appropriate for this purpose and we confirm that we considered to perform IHC for exactly this reason. Eventually, we however decided against performing additional stainings due to the associated additional financial costs and the good performance of automated tumour segmentation based on HE stained slides.

We would like to point out that before deciding to rely on whole-slide training annotations, we did run an additional AI training and analysis test round, with the following adaptions: We used n=23 training slides where training labels (i.e. BRAF positive or negative) were only assigned to neoplastic epithelium (instead of whole tissue section) and training/analysis magnification was set to 10x (instead of 2x). Interestingly, this approach did not improve the overall performance of the AI prediction. And in addition, extra time-consuming tumour segmentation had to be performed for the training slides. We therefore decided to continue with the whole-slide training annotations as indicated in the present study.

In addition, tumor location, grade, progression, and tissue composition should also be considered.

Yes, we agree that all these features are relevant for AI performance. To investigate the role of tissue composition, we primarily evaluated the tumour stroma for this feature. In particularly, we classified the samples based on the presence of stromal inflammation, which was confirmed to be relevant for the AI analysis. For this study, tumour grade was not considered a meaningful classifier as most cases were of high grade which is typical for canine UC. However, through the analysis of specific characteristics which define the tumour grade, such as quantitative evaluation of cellular and nuclear pleomorphism [11] would however enable further subclassifying tumours, even if they are of the same (i.e. high) grade. Such quantitative analysis was considered outside of the scope of this present study and was therefore not performed. We can however confirm that we are currently performing more detailed quantitative investigations in order to define relevant quantifiable features which define BRAF mutated canine UC. We aim to publish these results in a separate study. Regarding tumour location and progression, this information was unfortunately not available for most cases and was therefore not considered as a classifying criterium in the present study.

[1]       M. Haghighat, L. Browning, K. Sirinukunwattana, S. Malacrino, N. Khalid Alham, R. Colling, Y. Cui, E. Rakha, F.C. Hamdy, C. Verrill, J. Rittscher, Sci. Rep. 12 (2022) 1–16.

[2]       M. Salvi, U.R. Acharya, F. Molinari, K.M. Meiburger, Comput. Biol. Med. 128 (2021) 104129.

[3]       G. Campanella, M.G. Hanna, L. Geneslaw, A. Miraflor, V. Werneck Krauss Silva, K.J. Busam, E. Brogi, V.E. Reuter, D.S. Klimstra, T.J. Fuchs, Nat. Med. 25 (2019) 1301–1309.

[4]       R.H. Kim, S. Nomikou, N. Coudray, G. Jour, Z. Dawood, R. Hong, E. Esteva, T. Sakellaropoulos, D. Donnelly, U. Moran, A. Hatzimemos, J.S. Weber, N. Razavian, I. Aifantis, D. Fenyo, M. Snuderl, R. Shapiro, R.S. Berman, I. Osman, A. Tsirigos, J. Invest. Dermatol. 142 (2022) 1650-1658.e6.

[5]       Y. Terada, T. Takahashi, T. Hayakawa, A. Ono, T. Kawata, M. Isaka, K. Muramatsu, K. Tone, H. Kodama, T. Imai, A. Notsu, K. Mori, Y. Ohde, T. Nakajima, T. Sugino, T. Takahashi, JCO Clin. Cancer Informatics (2022) 1–14.

[6]       C. Nero, L. Boldrini, J. Lenkowicz, M.T. Giudice, A. Piermattei, F. Inzani, T. Pasciuto, A. Minucci, A. Fagotti, G. Zannoni, V. Valentini, G. Scambia, Int. J. Mol. Sci. 23 (2022).

[7]       R. Yamashita, J. Long, T. Longacre, L. Peng, G. Berry, B. Martin, J. Higgins, D.L. Rubin, J. Shen, Lancet Oncol. 22 (2021) 132–141.

[8]       M. Chen, B. Zhang, W. Topatana, J. Cao, H. Zhu, S. Juengpanich, Q. Mao, H. Yu, X. Cai, Npj Precis. Oncol. 4 (2020) 1–7.

[9]       S. Wu, G. Hong, A. Xu, H. Zeng, X. Chen, Y. Wang, Y. Luo, P. Wu, C. Liu, N. Jiang, Q. Dang, C. Yang, B. Liu, R. Shen, Z. Chen, C. Liao, Z. Lin, J. Wang, T. Lin, Lancet Oncol. 24 (2023) 360–370.

[10]     Y. Jiang, X. Sui, Y. Ding, W. Xiao, Y. Zheng, Y. Zhang, Front. Oncol. 12 (2023) 1–13.

[11]     R.H. Kim, S. Nomikou, N. Coudray, G. Jour, Z. Dawood, R. Hong, E. Esteva, T. Sakellaropoulos, D. Donnelly, U. Moran, A. Hatzimemos, J.S. Weber, N. Razavian, I. Aifantis, D. Fenyo, M. Snuderl, R. Shapiro, R.S. Berman, I. Osman, A. Tsirigos, J. Invest. Dermatol. 142 (2022) 1650-1658.e6.

Reviewer 2 Report

Thank you very much for this very interesting manuscript. In fact, I have no comments to add to that.

Author Response

We thank the reviewers for their comments. All comments of the four reviewers have been addressed and the manuscript has been changed accordingly. All performed changes are indicated by yellow highlight in the revised manuscript. The reviewers' comments are listed below, written in bold, followed by the authors' answers.

Reviewer 3 Report

An interesting study into the use of AI in predicting BRAF mutation status is presented. The study certainly deserves publication, although some remarks can be made.

Please change references notifications everywhere from [1][3-5][7] into [1, 3-5,7].

Line78-80: Ref 26 is not about testing but is about how many of the UT carcinomas are BRAF positive by PCR of tumor tissue.

Line 128-130: vague classification of inflammation. Please indicate more objectively description of marked inflammation.

177 samples of 157 dogs/tumors were used. Some tumors therefore have more influence on outcome than others. Wasn’t more appropriate to have of each tumor only one (the best) sample?

Line 229: well-preserved tissue in 53 out of 96 (Table 1): this info is not presented in Table 1.

Line 228-239: Unclear is which cases are being discussed: those of the training set or of the validation set or combined.

Line 240: “After initial histomorphologic examination, the designed AI tool was used to predict BRAF mutation on all available (n=97) WSI”  The validation set had 96 slides.

Line 253: Considering the relative low numbers please give 95% CI of the sensitivity and specificity. Also for other sensitivities and specificities listed further down in manuscript.

Line 298-302 (heading Table 4): info on statistics is part of M&M and can be removed here.

Line 311-315: is this negative BRAF prediction including the false negatives? It would be more interesting to see what are the criteria that are correlated with a true positive and true negative prediction.

Line 491-195: “The present study demonstrates that AI is able to predict BRAF mutation status in canine bladder urothelial carcinomas even though with limitations which we were able to define. Important potential confounding factors are sample size and quality as well as tumour histomorphology. Once optimized for these features, AI is able to reliably predict BRAF mutation, to detect intratumoral mutation heterogeneity and to differentiate between malignant and benign urothelium.”  With an overall sensitivity of 58% and a specifity of 63% this statement is a bit too optimistic. This study demonstrates that the technique of AI is promising, but that further improvements are necessary as is the need of good biopsies. The Last line of the abstract gives a better description: This is the first study to demonstrate the  use of AI histology to predict BRAF mutation status in canine UC. Despite certain limitations, the results highlight the potential of AI in predicting molecular alterations on routine tissue sections.

Author Response

Please change references notifications everywhere from [1][3-5][7] into [1, 3-5,7].

This has been changed throughout the entire manuscript.

Line78-80: Ref 26 is not about testing but is about how many of the UT carcinomas are BRAF positive by PCR of tumor tissue.

This reference has been replaced[12] (line 79).

Line 128-130: vague classification of inflammation. Please indicate more objectively description of marked inflammation.

This has been clarified in the manuscript (lines 130-132).

177 samples of 157 dogs/tumors were used. Some tumors therefore have more influence on outcome than others. Wasn’t more appropriate to have of each tumor only one (the best) sample?

Thank you for commenting on this matter. We can confirm that for eight dogs, we have used multiple slides for two main reasons: i) the extent of benign vs. malignant tissue or ii) the tumour histmorphology (i.e. tumour subgross or cellular morphology, invasion, inflammation) was considered significantly different between the different slides. Due to the significant slide differences and low number (n=8) of dogs with multiple slides, we consider it unlikely that this approach influences the outcome of this analysis.

This has been clarified in the manuscript (lines 118-120).

Line 229: well-preserved tissue in 53 out of 96 (Table 1): this info is not presented in Table 1.

This sentence has been adjusted (lines 230-231).

Line 228-239: Unclear is which cases are being discussed: those of the training set or of the validation set or combined.

This has been clarified in the manuscript (line 230).

Line 240: “After initial histomorphologic examination, the designed AI tool was used to predict BRAF mutation on all available (n=97) WSI”.  The validation set had 96 slides.

This has been corrected (line 244).

Line 253: Considering the relative low numbers please give 95% CI of the sensitivity and specificity. Also for other sensitivities and specificities listed further down in manuscript.

Thank you for pointing this out. We have assessed 95% CI with NCSS 2022 (v22.04). In our opinion, the table becomes rather confusing with the 95% CI. We would therefore prefer to use the original table (and at best integrate the table with the CI as supplementary).

If you insist, we would of course comply with your request and publish it in this way.

You will therefore find the original table in the revised manuscript and a table with your proposal and 95% CI in the supplementary word document.

Line 298-302 (heading Table 4): info on statistics is part of M&M and can be removed here.

This has been adjusted (revised Table 4).

Line 311-315: is this negative BRAF prediction including the false negatives? It would be more interesting to see what are the criteria that are correlated with a true positive and true negative prediction.

Thank you for pointing this out. We can confirm that this statement refers to true as well as false negative predictions. As described in lines 206-212 of the originally submitted manuscript (revised manuscript lines 208-214), criteria for true predictions (negative and positive combined) were defined and their relevance evaluated statistically (Chi-square test was performed for categorical data, i.e. urothelial differentiation, inflammation, slide quality, invasive growth; 2-sided t-test for continuous data, i.e. sample size). As recommended, the same tests have now additionally been performed separately for true positive as well as true negative predictions.

Similar to Table 3, we believe that the additional p-values are rather confusing for the reader. We would therefore prefer to publish the original table and, if necessary, add the additional p-values in the form of a supplementary.

However, if you insist, we will of course publish the table with all additional values.

You will find the version with p-values for true positive and true negative in the supplementary word document (similar to table 3).

Line 491-195: “The present study demonstrates that AI is able to predict BRAF mutation status in canine bladder urothelial carcinomas even though with limitations which we were able to define. Important potential confounding factors are sample size and quality as well as tumour histomorphology. Once optimized for these features, AI is able to reliably predict BRAF mutation, to detect intratumoral mutation heterogeneity and to differentiate between malignant and benign urothelium.”  With an overall sensitivity of 58% and a specifity of 63% this statement is a bit too optimistic. This study demonstrates that the technique of AI is promising, but that further improvements are necessary as is the need of good biopsies. The Last line of the abstract gives a better description: This is the first study to demonstrate the  use of AI histology to predict BRAF mutation status in canine UC. Despite certain limitations, the results highlight the potential of AI in predicting molecular alterations on routine tissue sections

This has been corrected in the manuscript (lines 506-508).

References:

[12]     A.M. Rasteiro, E. Sá E Lemos, P.A. Oliveira, R.M. Gil da Costa, Vet. Sci. 9 (2022).

Reviewer 4 Report

This manuscript presents an interesting research on the possibilities of AI in veterinary medicine. The manuscript describes in detail Material and Methods  and clearly presents the results. I have no objections to the text of the manuscript.

Author Response

(The authors gave the same response as above.)
